# Stability of Nine Time-Dependent Antibiotics for Outpatient Parenteral Antimicrobial Therapy (OPAT) Use

**DOI:** 10.3390/antibiotics14050466

**Published:** 2025-05-03

**Authors:** Elise d’Huart, Ibtissem Boutouha, Clara Berardi, Jean Vigneron, Béatrice Demore, Alexandre Charmillon

**Affiliations:** 1Pharmacy Department, CHRU-Nancy, F-54000 Nancy, France; ibtissem.boutouha@chr-metz-thionville.fr (I.B.); c.berardi@chru-nancy.fr (C.B.); b.demore@chru-nancy.fr (B.D.); 2Infostab—A Nonprofit Association, F-54180 Heillecourt, France; vigneron.j@wanadoo.fr; 3Inserm, INSPIIRE, Universite de Lorraine, F-54000 Nancy, France; 4Infectious Diseases Department, CHRU-Nancy, F-54000 Nancy, France; a.charmillon@chru-nancy.fr

**Keywords:** stability, antibiotic, outpatient parenteral antimicrobial therapy, elastomeric diffuser

## Abstract

**Background:** The use of an elastomeric diffuser is favored to administer outpatient antibiotic therapy. A study published in 2022 highlighted the instability of several antibiotics in elastomeric devices at 37 °C. The objective was to evaluate the stability of nine time-dependent antibiotics that are unstable at 37 °C at lower concentrations and a reduced storage temperature of 32 °C. **Methods:** Chemical stability was assessed by pH measurement and high-performance liquid chromatography. Physical stability was evaluated by visual and subvisual inspection. The solutions were considered stable if the remaining drug percentage was ≥90%, the maximum variation in pH was less than 1, the particle count was within acceptable limits and the visual aspect remained unchanged after storage. **Results:** Solutions showing stability for 24 h are composed of 12.5 mg/mL cefiderocol in NS (normal saline) and 50–133 mg/mL piperacillin in NS-D5W (5% dextrose). Additionally, 12.5 mg/mL amoxicillin in NS; 12.5 mg/mL cefepime in NS-D5W; 12.5 mg/mL cefiderocol in D5W; 25 mg/mL cefiderocol in NS-D5W; 12.5 mg/mL cefotaxime in NS-D5W; 12.5 mg/mL cefoxitin in NS-D5W; 12.5 mg/mL ceftazidime in NS-D5W; 25 mg/mL ceftazidime in NS; 25 mg/mL cloxacillin in NS-D5W; and 25–50 mg/mL oxacillin in NS were shown to be stable for 12 h. Notably, 25 mg/mL amoxicillin in NS, 50 mg/mL cloxacillin in NS and 25 mg/mL oxacillin in D5W were shown to be stable for 8 h. **Conclusions:** These 12–24 h stability data indicate that these antibiotics can be administered by continuous infusion using only one–two elastomeric devices per day, facilitating outpatient parenteral antimicrobial therapy (OPAT).

## 1. Introduction

The continuous infusion of time-dependent antibiotics such as beta-lactams has been studied for several years, which has allowed optimization of therapeutic outcomes. The literature review by Diamantis et al. compared continuous or extended infusion of beta-lactams to their intermittent infusion. Clinical pharmacokinetic/pharmacodynamic (PK/PD) data have confirmed that continuous infusion of antibiotics be administered to severely ill patients or those infected by a pathogen with a high minimum inhibitory concentration (MIC), as continuous infusion is associated with increased survival rates. Furthermore, continuous infusion may allow for a decrease in the daily dosage of antibiotics, thereby contributing to lower overall costs [1].

For several years, overburdened hospitals have encouraged the use of outpatient parenteral antibiotic therapy (OPAT) whenever possible. The continuous infusion of time-dependent antibiotics such as beta-lactams allows better treatment compliance, fewer nursing visits and improved therapeutic outcomes. The major benefits of OPAT are the need for fewer or no hospital stays, a reduced risk of nosocomial infection, significant cost savings, and an improved quality of life for the patient [2,3]. Elastomeric devices, which allow continuous infusions to be performed in the patient’s home, are preferred for OPAT, as they are mechanical devices that require neither gravity nor electricity to operate [4].

In 2022, Loeuille et al. published an article summarizing stability studies of 16 time-dependent antibiotics to promote their continuous administration either via syringes in intensive care units or elastomeric diffusers for OPAT [5]. This study highlighted the instability of several antibiotics in elastomeric diffusers at 37 °C, such as 50 mg/mL cefepime in normal saline (NS), 25 mg/mL cefotaxime in NS and 5% dextrose (D5W), 25 mg/mL cefoxitin in NS and D5W, 50–100 mg/mL cloxacillin in NS and D5W, and 66.7 mg/mL piperacillin in NS and D5W.

Additionally, the physicochemical stability of 25 mg/mL cefiderocol in NS or D5W is limited to 6 h, necessitating four administrations a day to ensure continuous infusion, i.e., four nursing rounds. Other solutions were unstable after 24 h, but their stability was not assessed at 8 h or 12 h. The stability of 25 mg/mL ceftazidime was evaluated after 8 h and 24 h of storage at 37 °C but not after 12 h. Additional solutions were stable at 8 h but unstable at 24 h. Dhelens et al. studied the stability of 50 mg/mL oxacillin in NS and D5W in elastomeric devices, and a variation in nearly 1 pH unit was observed after 12 h of storage at 35 °C [6].

Due to the lack of sufficient stability data for these antibiotics in elastomeric devices, they cannot be continuously infused using these devices; thus, OPAT cannot be optimized.

Various factors, such as the storage temperature or concentration, can influence the stability of a molecule [7]. Van Der Merwe et al. suggested evaluating stability in elastomeric devices at a temperature of 30 °C [8]. The standard storage temperature in elastomeric devices recommended by the National Health Service (NHS) is 32 °C ± 1 °C [9]. The NHS also stated that the temperature of these solutions should not exceed 32 °C whenever possible.

The aim of this study was to investigate the stability of nine time-dependent antibiotics previously demonstrated to be unstable at 37 °C. Amoxicillin, cefepime, cefiderocol, cefotaxime, cefoxitin, ceftazidime, cloxacillin, oxacillin and piperacillin solutions were stored in silicone elastomeric devices at 32 °C at lower concentrations than those reported in the literature.

## 2. Results

### 2.1. Chemical Stability Determined by High-Performance Liquid Chromatography (HPLC)

The complete, validated analytical HPLC method to determine antibiotic stability was reported by Loeuille et al. (except for oxacillin, which was not studied by Loeuille et al., and cloxacillin) and was evaluated for linearity, precision, and specificity. The results from the reevaluation of linearity and precision for all antibiotics are presented in Table 1.

Oxacillin and cloxacillin demonstrated linearity, with R^2^ values of 0.99983 and 0.99980, respectively. The calibration curves were linear according to nonlinear ANOVA (F_exp_ < Fth = 3.71). The homogeneity of the variances was also proven by a Cochran test (C_exp_ < Cth = 0.684). Accuracy has been validated for each method and values of limit of detection (LOD) and limit of quantification (LOQ) are presented in Table 1. The intraday and interday precisions were less than 2% for oxacillin and cloxacillin. Thus, the analytical method was stable for these two antibiotics.

Table 2 presents the concentrations of antibiotics in the elastomeric devices at various times. Table 3 presents the amount of pyridine detected during the ceftazidime stability study.

### 2.2. pH Measurements

The pH measurement results are presented in Table 4. The pH varied by less than one pH unit for all tested samples except for the following samples: 12.5 mg/mL cefoxitin in NS after 24 h (variation of 1.22 pH units); cloxacillin under all the conditions tested (minimum variation of 1.0 pH unit and maximum variation of 1.52 pH units); and 25 mg/mL oxacillin in NS (variation of 1.0 pH unit) and D5W (variation of 1.25 pH units). However, the pH values of these samples were consistent after 12 h. For 50 mg/mL oxacillin, the change in pH approached the limit at 24 h (variation of 0.92 pH units in NS and 0.96 in D5W).

### 2.3. Visual and Subvisual Evaluations

The visual and subvisual evaluation results are presented in Table 4. Yellowing of the preparations was observed during our stability studies, except for 50 mg/mL cloxacillin in D5W, 25 mg/mL oxacillin in D5W and 50 mg/mL oxacillin in both solvents studied.

The particle counter was compliant for most samples in this work, except for 50 mg/mL oxacillin in D5W, which was not compliant at 8 h.

### 2.4. Control Experiments

Regarding the ED of NS and D5W alone, visual examination was compliant (no change in color, precipitate or gas formation) for 24 h; subvisual examination was compliant with European Pharmacopoeia monograph 2.9.19. The change in pH was also less than 0.18 units for 24 h.

### 2.5. Summary of the Results

On the basis of the physical and chemical test results, a stable duration was established for each preparation (antibiotic, concentration, solvent, and storage temperature of 32 °C). A summary of the stability data obtained is presented in Table 5.

## 3. Discussion

These studies yielded data regarding the stability of time-dependent antibiotics that are unstable at 37 °C in a silicone elastomer device at 32 °C. These data will enable continuous antibiotic infusion at home. For intensive care units (ICUs), stability studies in syringes are necessary and are presented in the work by Loeuille et al. [5].

### 3.1. Temperature Selection

Loeuille et al. evaluated antibiotic stability at body temperature to consider the worst-case scenario. The antibiotics studied here were shown to be unstable at 37 °C or had limited data that could not be used for clinical application. Temperature affects antibiotic stability [7]. In this study, the storage temperature of the antibiotics in the elastomeric device was reduced to 32 °C, as recommended by the NHS [9]. It is important that patients be educated to ensure safe and effective drug administration and that the temperature of the portable device does not exceed that recommended by the stability studies. Inside the elastomeric device, the temperature of the solution can vary depending on how the device is placed: protected or not from light, under the patient’s clothes or not. The use of an isothermal bag would reduce this temperature variation, but after research with our suppliers of elastomeric dispensers, this type of bag did not exist.

### 3.2. Assessment of Additional Peak

#### 3.2.1. Production During the Ceftazidime Stability Study

Pyridine is one of the ceftazidime degradation products that may be toxic. Pyridine toxicity and its toxic concentration have been the subject of debate for several decades [10,11,12,13,14]. Pyridine is considered a central nervous system depressant in humans and can cause liver, kidney and digestive disorders [15,16]. Jones et al. suggested that a safe dosage of pyridine is 100 mg per day [17]. Under all the conditions tested during this work, the pyridine level increased progressively over time during ceftazidime storage, but less than 100 mg accumulated in all cases. Ceftazidime (25 mg/mL) in D5W gave the highest concentration of pyridine. Given the doses of pyridine measured in our stability study, the risk seems minimal.

#### 3.2.2. Other Antibiotics

For each stability study, we have studied the additional peaks present on the chromatograms at each analysis time. Their evolution during the analyses was determined as well as their presence in the analysis of a quality sample. The percentage of AUC of additional peak/AUC of antibiotic of interest was measured for each antibiotic and was less than 2% for each antibiotic. Additional peaks that are present after preparation (T0) and evolve constantly over time, or additional peaks that are also present on the chromatogram of the CRS analysis, are considered impurities. A literature search for potential degradation products (DPs) for each antibiotic was also carried out. To our knowledge, with the exception of ceftazidime with pyridine, no study demonstrated the toxicity of a PD for an antibiotic studied in this study.

### 3.3. Solvents Used

NS and D5W were used as the solvents for all samples. Antibiotic stability can differ in different solvents depending on the pH (D5W has a pH of approximately 4 and NS has a pH of approximately 6–7) or the presence of chloride ions in NS. Some products are not stable in either solvent. Thus, an appropriate solvent should be chosen on the basis of the stability study data. Loeuille et al. discussed the use of citrate buffer to obtain a pH close to 7, which has been previously reported and applied in the UK to improve the stability of unstable molecules. In France, using buffer as a solvent is not possible in clinical practice, which restricts the application of this approach.

### 3.4. Limiting Factors

#### 3.4.1. Container Type

The stability studies performed in elastomeric diffusers in this study were carried out using silicone devices. Other diffusers, such as those composed of polyisoprene, are also available. We chose silicone diffusers due to their availability in our hospital. However, we do not recommend extrapolating data from silicone diffusers for application to polyisoprene diffusers. The results obtained in this work are only available for the preparation of silicone elastomer dispensers and may not apply to polyisoprene devices.

#### 3.4.2. pH Value

The acceptability criterion for pH measurements was a maximum variation of 1 pH unit. pH was a factor limiting the stability of some of our samples. For 25 mg/mL cloxacillin in NS and D5W and for 50 mg/mL cloxacillin in NS, the HPLC and visual and subvisual examination results met the acceptability criteria, but the pH varied by more than one unit, limiting the stability of the sample.

#### 3.4.3. Visual and Subvisual Examinations

Loeuille et al. highlighted that the physical appearance of the tested solutions in several stability studies changed, manifested by yellowing of the preparations. This visual phenomenon was also observed for some of the samples in our stability study. Loeuille et al. reported visual changes for 50 mg/mL cefepime in NS, 25 mg/mL cefotaxime in NS and D5W, 25 mg/mL cefoxitin, 50 and 100 mg/mL cloxacillin and 66.7 mg/mL piperacillin. These samples also exhibited changes under some conditions (Table 4) in our study, even though the concentrations were reduced and the storage temperature was lower (32 °C). The visual examination was a factor that limited the stability 12.5 mg/mL cefepime in NS and D5W and 25 mg/mL oxacillin in D5W, while all the other parameters of these samples were consistent at the next analysis time.

A particle counter was used during each stability analysis to measure the number of particles with sizes greater than 10 and 25 µm according to the European Pharmacopeia protocol. These criteria were factors limiting the stability of 25 mg/mL cefiderocol in NS, as only the subvisual examination data was out of specification for this sample.

This study complements that of Loeuille et al. and states the conditions for preparing solutions for continuous infusion to optimize the administration of beta-lactam antibiotics via OPAT.

#### 3.4.4. Effects of Concentration, Solvent and Temperature

Amoxicillin was stable for 12 h at a concentration of 12.5 mg/mL and for 8 h at a concentration of 25 mg/mL but unstable after 8 h at a concentration of 50 mg/mL. Therefore, the solution concentration influences the stability of amoxicillin. Cloxacillin was more stable at 12.5 mg/mL than at 25 mg/mL; thus, the solution concentration also influences the stability of cloxacillin. The concentration of the oxacillin solution also affected stability, as 25 and 50 mg/mL oxacillin was stable for 12 h in NS 25 mg/mL oxacillin was stable for 8 h in D5W, but 50 mg/mL oxacillin was unstable in D5W. In D5W, significant yellowing of the oxacillin solution was observed, limiting its stability.

Loeuille et al. demonstrated that 66.7 mg/mL piperacillin was unstable in NS and D5W at 37 °C. However, piperacillin was stable for up to 24 h at 32 °C at a higher concentration than those described by Loeuille et al.

## 4. Materials and Methods

### 4.1. Chemicals, Reagents and Products Used

The reagents used for mobile phase preparation and during analytical method validation were of HPLC grade. Ultrapure water for chromatography was obtained using an ELGA Purelab Flex water purification system (ELGA LabWater, Woodridge, IL, USA). The antibiotics used for solution preparation are summarized in Table 6. The normal saline (NS), sterile water for injection (SWFI) and 5% dextrose (D5W) solvents used to reconstitute or dissolve the drugs were purchased in glass vials (Chaix et du Marais, Lavoisier, Paris, France). The prepared drug solutions were stored in silicone elastomeric devices (Autofuseur 10 mL/h, 275 mL; ACE Medical, AA2010-1-S; batch: A2301414, Goyang-si, Republic of Korea).

### 4.2. Instruments

The following instruments were used for the stability studies:The high-performance liquid chromatography (HPLC) system consisted of an ELITE LaChromVWR/Hitachi Plus autosampler (Kawasaki-shi, Japan), a photodiode array detector (VWR, L-2455, Hitachi) and an HPLC pump (VWR, L-2130, Hitachi). Data were acquired and analyzed by using EZChrom Elite (VWR, Agilent, Santa Clara, CA, USA).pH meter (Hanna Edge, Smithfield, RI, USA).PAMAS particle counter (Stuttgart, Germany).

### 4.3. Methods

#### 4.3.1. Test Solution Preparation and Storage

The drug concentrations, solvents and analysis times were chosen on the basis of the results of Loeuille et al. and collegial decisions between an infectious disease specialist and a pharmacist considering observed practices [5]. Elastomeric preparations were made in elastomeric devices with a total volume of 120 or 240 mL.

The preparations of the solutions tested are presented in Table 7.

The elastomeric devices were stored at 32 °C in a climatic chamber in the dark. The stability of antibiotics was assessed at different times over 24 h. Three elastomeric devices were prepared for each treatment condition.

To expedite analysis, a 24 h prestudy was performed in glass vials and elastomeric devices at 32 °C for all antibiotics except cefiderocol, cloxacillin and oxacillin, in which the concentrations and solvents varied. The aim of this preliminary study was to visually evaluate the physical stability of the different solutions using a particle counter to identify those in which precipitates formed or that changed color. On the basis of the results obtained, various conditions were selected for the elastomeric diffuser study. Because glass is more inert than silicone, if physical instability was observed in glass, than that condition was not selected for the complete physicochemical stability study in the diffuser. In addition, this prestudy enabled us to reduce costs and save time by selecting only the most relevant conditions. The chemical stability of amoxicillin at 6 h and 24 h was also evaluated by HPLC in the prestudy.

#### 4.3.2. HPLC Analysis of Chemical Stability

The stability of the antibiotic solutions was analyzed by reversed-phase HPLC according to methods in the literature. The analytical HPLC conditions (including the mobile phase composition, pH, flow rate, injection volume, detection wavelength, retention time and reference) are presented in the Appendix A, except for those used for oxacillin and cloxacillin analysis. Three samples were prepared for each condition and analyzed by HPLC at each analysis time.

The stability of the oxacillin and cloxacillin solutions was analyzed by reversed-phase high-performance liquid chromatography (RP-HPLC) and photodiode array detection [18]. The mobile phase consisted of acetonitrile (VWR Chemicals) and ultrapure water, and the pH was adjusted to 3 with orthophosphoric acid (VWR Chemicals, batch: 21L204004). The flow rate was 1.2 mL/min, and the injection volume was 10 µL. The detection wavelength was 270 nm. The temperature of the injector was 10 °C, and that of the column oven was 20 °C. Calibration curves were constructed by plotting the peak area as a function of the concentration. The linearity of the method was evaluated using five concentrations of standard solutions.

The ceftazidime molecule has a pyridine ring which is released during hydrolysis. It is one of the major degradation products of ceftazidime but is considered toxic. For the ceftazidime stability study, the pyridine concentration was also determined at each analysis time. Jones et al. suggested that a safe dose of pyridine is 100 mg per day [17].

The antibiotics were considered chemically stable if not less than 90% of the initial concentration remained considering the formation of degradation products [19,20]. The results presented are an average of the three preparations made for each condition.

#### 4.3.3. Analytical Method Validation

Analytical method validation was performed as recommended by the International Conference on Harmonisation Q2R1 (ICH) and as described by Loeuille et al. For each antibiotic (except oxacillin and cloxacillin), linearity was evaluated by analyzing three sets of five concentrations of standard solutions over three days, and these data were also used to evaluate method precision (intraday reproducibility and interday precision). For accuracy, three different solutions of three concentrations were prepared three times a day for 3 days for each antibiotic. Accuracy was determined as the difference between the mean measured value and the accepted true value. For each antibiotic, recovery at three concentration levels must be between 98 and 102% to validate accuracy. During the ceftazidime stability study, 5 concentrations of the pyridine standard (1–40 µg/mL) were used to determine the amount of pyridine generated at each analysis point.

For oxacillin and cloxacillin, complete validation of the analytical method was carried out. Calibration curves were constructed by plotting the peak area as a function of concentration. The linearity of the method was evaluated using 5 concentrations of the standard. The homogeneity of variance of the calibration curves was evaluated with a Cochran test, and a value of *p* < 0.05 was considered to indicate statistical significance. The linear regression data were subjected to analysis of variance (ANOVA) to determine significance (*p* < 0.05). The intraday reproducibility and interday precision were also determined. Selectivity and specificity were evaluated by forced degradation studies under acidic, basic, thermal, photolytic and oxidative conditions. The objective was to degrade between 10% and 20% of the molecule of interest to prove that the methods were stable [7].

#### 4.3.4. Physical Stability

Antibiotics were considered physically stable when particulates, haze formation and color change were not observed. The samples were visually inspected with the unaided eye against a white/black background at each analysis time. Physical stability was also assessed by performing a particulate contamination test (PAMAS SVSS) at the beginning and end of the study. The results were analyzed according to the criteria of the European Pharmacopoeia [21].

#### 4.3.5. pH Measurements

The pH of each solution was measured at each analysis time. A variation of more than one pH unit was considered unacceptable [7].

#### 4.3.6. Control Experiments

To check that the pH and physical changes were not related to the solvent but to the antibiotic itself, three EDs of NS and three EDs of D5W were prepared and stored at 32 °C during 24 h. pH assessment, visual examination and subvisual examination were performed for 24 h.

## 5. Conclusions

Among the nine antibiotics tested for stability at 32 °C, 50 and 133 mg/mL piperacillin in NS and D5W and 12.5 mg/mL cefiderocol in NS were stable for 24 h, necessitating only one nurse visit per day. Solutions that are stable for 12 h, including amoxicillin (12.5 mg/mL in NS), cefepime (12.5 mg/mL in NS and D5W), cefiderocol (12.5 mg/mL in D5W and 25 mg/mL in NS and D5W), cefotaxime (12.5 mg/mL in NS and D5W), cefoxitin (12.5 mg/mL in NS and D5W), ceftazidime (12.5 mg/mL in NS and D5W and 25 mg/mL only in NS), cloxacillin (25 mg/mL in NS and D5W), and oxacillin (25 and 50 mg/mL in NS), necessitate only two nurse visits per day.

This study presents new stability data determined upon reducing the concentrations of the solutions and lowering the storage temperature of the dispensers to 32 °C to enable continuous infusion for OPAT. Importantly, the stability of a medication in a syringe is not the same as that in a dispenser. An appropriate solvent should also be chosen on the basis of these stability data because stability in NS and D5W differed significantly. It is also important to educate patients on the proper storage temperature (32 °C) when a portable dispenser is chosen as the delivery device.

These new stability data allow the optimization of the PK/PD parameters of β-lactam antibiotics and ensure that continuous infusion is safe. These data will enable nurse coordinators to reduce the number of visits to each patient’s home to change the portable dispenser. The β-lactams investigated were stable for at least 12 h under the conditions studied, which will facilitate OPAT and allow only two nursing visits to the patient’s home each day. The nurses should also be trained on how to properly implement and administer these medications using portable dispensers.

This study complements the work of Loeuille et al. (2022) [5] by proposing methods to provide antibiotic therapy at home using portable dispensers.

These new stability data indicate that it is possible to safely administer antibiotics continuously via silicone elastomeric dispensers in the patient’s home. These stability data suggest that educating patients to control the temperature of the device at 32 °C is critical, as the dispenser should not be placed near the body overnight, as maintaining the proper temperature is essential. It might be advisable to place the device in an isothermal bag to maintain the temperature at a maximum of 32 °C.

## Figures and Tables

**Table 1 antibiotics-14-00466-t001:** Validation criteria for the analytical HPLC method.

Antibiotic	Calibration Range (µg/mL)	R^2^	Intraday Precision [min; max] (%)	Interday Precision [min; max] (%)	Cochran’s Test C_exp_ (*p* < 0.05)	ANOVA (Nonlinear) F_exp_ (*p* < 0.05)	Recovery–Accuracy %	LOD (Limit of Detection)/LOQ (Limit of Quantification) (µg/mL)
Amoxicillin	120–280	0.999	[0.16; 1.73]	[0.83; 3.42]	0.266	0.03	[99.97–100.17%]	1.79/5.43
Cefepime	60–140	0.999	[0.12; 1.92]	[1.19; 2.20]	0.339	0.32	[98.41–100.39%]	3.00/9.08
Cefiderocol	25–75	0.999	[0.17; 1.89]	[1.89; 3.79]	0.519	0.06	[99.48–100.25%]	0.71/2.16
Cefotaxime	50–150	0.999	[0.66; 1.74]	[1.18; 2.31]	0.327	0.02	[99.56–100.65%]	1.35/4.09
Cefoxitin	75–175	0.999	[0.02; 1.00]	[1.78; 2.94]	0.391	0.05	[99.35–100.36%]	1.54/4.68
Ceftazidime	100–500	0.999	[0.28; 1.49]	[1.60; 3.08]	0.452	0.33	[99.24–100.80%]	8.13/24.65
Cloxacillin	30–70	0.999	[0.04; 1.78]	[0.71; 1.21]	0.432	0.10	[99.45–101.05%]	1.04/3.15
Oxacillin	30–70	0.999	[0.04; 1.50]	[0.44;1.73]	0.398	0.18	[99.89–100.76%]	0.97/2.94
Piperacillin	100–300	0.999	[0.41; 1.41]	[1.14; 1.99]	0.659	1.39	|96.62–100.84%]	12.12/36.73

**Table 2 antibiotics-14-00466-t002:** Stability of antibiotics in silicone elastomeric devices at 32 °C for OPAT use.

				Mean Percent of Initial Concentration ± RSD * (%)				
Antibiotic	Concentration	Solvent	Initial Concentration (mg/mL)	T0 h	T6 h	T8 h	T12 h	T24 h
Amoxicillin	12.5 mg/mL	NS **	12.17	100.0 ± 1.49	/	92.90 ± 0.76	91.52 ± 0.44	77.88 ± 2.24
	25 mg/mL	NS	23.13	100.0 ± 0.87	91.09 ± 0.51	91.94 ± 0.51	/	77.89 ± 2.25
	50 mg/mL	NS	48.47	100.0 ± 1.38	84.95 ± 1.80	/	/	46.78 ± 1.12
Cefepime	12.5 mg/mL	NS	12.12	100.0 ± 0.64	101.60 ± 1.01	97.96 ± 1.05	99.57 ± 0.54	92.55 ± 1.82
		D5W ***	11.81	100.0 ± 0.77	100.76 ± 0.50	100.93 ± 0.50	99.33 ± 0.85	94.64 ± 1.07
Cefiderocol	12.5 mg/mL	NS	11.97	100.0 ± 1.30	/	98.38 ± 0.87	100.80 ± 2.11	95.44 ± 1.77
		D5W	12.32	100.0 ± 0.62	/	96.80 ± 3.58	93.72 ± 1.93	89.26 ± 2.23
	25 mg/mL	NS	24.03	100.0 ± 1.24	/	100.01 ± 1.31	96.48 ± 2.35	91.16 ± 0.93
		D5W	24.22	100.0 ± 1.33	/	95.79 ± 2.03	94.75 ± 1.30	89.67 ± 2.09
Cefotaxime	12.5 mg/mL	NS	11.94	100.0 ± 1.30	/	98.25 ± 4.36	93.76 ± 1.20	79.47 ± 2.50
		D5W	11.67	100.0 ± 1.24	/	97.2 ± 1.24	97.07 ± 2.46	80.77 ± 1.18
Cefoxitin	12.5 mg/mL	NS	12.59	100.0 ± 2.18	98.66 ± 0.71	97.95 ± 0.38	94.78 ± 0.24	84.90 ± 1.51
		D5W	12.80	100.0 ± 1.35	95.65 ± 0.68	96.81 ± 0.19	95.31 ± 0.47	87.56 ± 0.26
Ceftazidime	12.5 mg/mL	NS	12.01	100.0 ± 1.95	/	/	93.14 ± 1.42	85.91 ± 1.86
		D5W	12.58	100.0 ± 4.82	/	/	97.32 ± 0.58	86.80 ± 0.77
	25 mg/mL	NS	23.32	100.0 ± 0.98	/	/	95.22 ± 1.59	69.40 ± 1.40
		D5W	29.48	100.0 ± 1.53	/	/	86.78 ± 1.59	75.06 ± 1.15
Cloxacillin	25 mg/mL	NS	25.13	100.0 ± 1.00	/	99.81 ± 2.26	101.02 ± 1.37	96.18 ± 1.24
		D5W	24.99	100.0 ± 1.00	/	100.78 ± 0.70	101.35 ± 1.06	99.16 ± 0.90
	50 mg/mL	NS	50.49	100.0 ± 2.40	/	98.62 ± 1.53	100.01 ± 1.39	95.09 ± 2.56
		D5W	49.33	100.0 ± 1.26	/	101.62 ± 2.00	102.85 ± 0.57	98.60 ± 1.27
Oxacillin	25 mg/mL	NS	23.91	100.0 ± 1.60	/	98.08 ± 1.99	97.93 ± 2.20	91.91 ± 1.99
		D5W	23.95	100.0 ± 0.83	/	100.12 ± 1.11	100.23 ± 0.79	95.42 ± 1.42
	50 mg/mL	NS	49.53	100.0 ± 0.89	/	96.34 ± 1.99	97.12 ± 1.86	88.62 ± 1.90
		D5W	49.45	100.0 ± 1.29	/	97.38 ± 1.10	98.79 ± 1.28	89.77 ± 1.13
Piperacillin	50 mg/mL	NS	48.01	100.0 ± 1.19	99.14 ± 0.52	99.32 ± 0.78	98.98 ± 0.67	95.39 ± 1.18
		D5W	50.58	100.0 ± 2.57	103.00 ± 1.11	99.77 ± 1.11	Technical problems	93.08 ± 1.64
	133 mg/mL	NS	126.49	100.0 ± 1.12	100.47 ± 0.95	100.97 ± 1.25	100.91 ± 0.74	96.33 ± 3.66
		D5W	124.40	100.0 ± 1.87	Technical problems	Technical problems	Technical problems	94.74 ± 1.73

* RSD: relative standard deviation; ** NS: normal saline; *** D5W: 5% dextrose; /: analysis not performed.

**Table 3 antibiotics-14-00466-t003:** Amount of pyridine detected during the ceftazidime stability study.

			Average Total Amount of Pyridine (mg)		
Antibiotic	Concentration	Solvent	T0 h	T12 h	T24 h
Ceftazidime	12.5 mg/mL	NS	0.78	11.22	25.76
		D5W	1.02	27.82	57.45
	25 mg/mL	NS	1.18	11.47	26.57
		D5W	2.02	42.93	76.50

**Table 4 antibiotics-14-00466-t004:** pH measurement and visual and subvisual examination results.

			Physical Stability		pH Measurements	Variation in pH (Absolute Value)
Antibiotic	Concentration	Solvent	Visual Evaluation	Subvisual Evaluation		
Amoxicillin	12.5 mg/mL	NS *	Compliant after 12 h	Compliant after 12 h	Compliant after 12 h	0.27 (T0–T24 h)
	25 mg/mL	NS	Compliant after 8 h	Compliant after 8 h	Compliant after 8 h	0.16 (T0–T8 h)
	50 mg/mL	NS	Compliant after 24 h	Compliant after 24 h	Compliant after 24 h	Not determined
Cefepime	12.5 mg/mL	NS	Compliant after 12 h	Compliant after 24 h	Compliant after 24 h	0.25 (T0–T24 h)
		D5W **	Compliant after 12 h	Compliant after 24 h	Compliant after 24 h	0.33 (T0–T24 h)
Cefiderocol	12.5 mg/mL	NS	Compliant after 24 h	Compliant after 24 h	Compliant after 24 h	0.24 (T0–T24 h)
		D5W	Compliant after 24 h	Compliant after 24 h	Compliant after 24 h	0.15 (T0–T24 h)
Cefiderocol	25 mg/mL	NS	Compliant after 24 h	Compliant after 12 h	Compliant after 24 h	0.07 (T0–T24 h)
		D5W	Compliant after 24 h	Compliant after 24 h	Compliant after 24 h	0.08 (T0–T24 h)
Cefotaxime	12.5 mg/mL	NS	Compliant after 24 h	Compliant after 24 h	Compliant after 24 h	0.51 (T0–T24 h)
		D5W	Compliant after 24 h	Compliant after 24 h	Compliant after 24 h	0.55 (T0–T24 h)
Cefoxitin	12.5 mg/mL	NS	Compliant after 12 h, yellowing at 24 h	Compliant after 24 h	Compliant after 12 h	1.22 (T0–T24 h) 0.55 (T0–T12 h)
		D5W	Compliant after 24 h	Compliant after 24 h	Compliant after 24 h	0.54 (T0–T24 h)
Ceftazidime	12.5 mg/mL	NS	Compliant after 24 h	Compliant after 24 h	Compliant after 24 h	0.61 (T0–T24 h)
		D5W	Compliant after 24 h	Compliant after 24 h	Compliant after 24 h	0.32 (T0–T24 h)
	25 mg/mL	NS	Compliant after 24 h	Compliant after 24 h	Compliant after 24 h	0.68 (T0–T24 h)
		D5W	Compliant after 24 h	Compliant after 24 h	Compliant after 24 h	0.38 (T0–T24 h)
Cloxacillin	25 mg/mL	NS	Compliant after 24 h	Compliant after 24 h	Compliant after 12 h	1.17 (T0–T24 h) 0.75 (T0–T12 h)
	25 mg/mL	D5W	Compliant after 24 h	Compliant after 24 h	Compliant after 12 h	1.00 (T0–T24 h) 0.66 (T0–T12 h)
	50 mg/mL	NS	Compliant after 24 h	Compliant after 24 h	Compliant after 8 h	1.52 (T0–T24 h) 0.93 (T0–T8h)
	50 mg/mL	D5W	Not compliant at 8 h (yellowing)	Compliant after 8 h	Compliant after 12 h	1.00 (T0–T24 h) 0.63 (T0–T12 h)
Oxacillin	25 mg/mL	NS	Compliant after 24 h	Compliant after 24 h	Compliant after 12 h	1.00 (T0–T24 h) 0.57 (T0–T12H)
	25 mg/mL	D5W	Compliant after 8 h	Compliant after 24 h	Compliant after 12 h	1.24 (T0–T24 h) 0.86 (T0–T12 h)
	50 mg/mL	NS	Compliant after 12 h	Compliant after 12 h	Compliant after 24 h	0.92 (T0–T24 h)
	50 mg/mL	D5W	Not compliant at 8 h (yellowing)	Not compliant at 8 h	Compliant after 24 h	0.96 (T0–T24 h)
Piperacillin	50 mg/mL	NS	Compliant after 24 h	Compliant after 24 h	Compliant after 24 h	0.66 (T0–T24 h)
		D5W	Compliant after 24 h	Compliant after 24 h	Compliant after 24 h	0.66 (T0–T24 h)
	133 mg/mL	NS	Compliant after 24 h	Compliant after 24 h	Compliant after 24 h	0.54 (T0–T24 h)

* NS: normal saline; ** D5W: 5% dextrose.

**Table 5 antibiotics-14-00466-t005:** Stability of the antibiotics in silicone elastomeric devices at 32 °C.

Antibiotic	Amount (g)	Solvent	Stability Duration in a Silicone Elastomeric Device at 32 °C (Hours)
	Concentration		
Amoxicillin	(3 g/240 mL) 12.5 mg/mL	NS *	12 h
	(6 g/240 mL) 25 mg/mL	NS	8 h
	(12 g/240 mL) 50 mg/mL	NS	Unstable at 8 h
Cefepime	(3 g/240 mL) 12.5 mg/mL	NS	12 h
		D5W **	12 h
Cefiderocol	(3 g/240 mL) 12.5 mg/mL	NS	24 h
		D5W	12 h
	(6 g/240 mL) 25 mg/mL	NS	12 h
		D5W	12 h
Cefotaxime	(3 g/240 mL) 12.5 mg/mL	NS	12 h
		D5W	12 h
Cefoxitin	(3 g/240 mL) 12.5 mg/mL	NS	12 h
		D5W	12 h
Ceftazidime	(3 g/240 mL) 12.5 mg/mL	NS	12 h
		D5W	12 h
	(6 g/240 mL) 25 mg/mL	NS	12 h
		D5W	Unstable at 12 h Stable at 8 h * (Loeuille et al.)
Cloxacillin	(6 g/240 mL) 25 mg/mL	NS	12 h
		D5W	12 h
	(12 g/240 mL) 50 mg/mL	NS	8 h
		D5W	Unstable at 8 h
Oxacillin	(6 g/240 mL) 25 mg/mL	NS	12 h
		D5W	8 h
	(12 g/240 mL) 50 mg/mL	NS	12 h
		D5W	Unstable at 8 h
Piperacillin	(12 g/240 mL) 50 mg/mL	NS	24 h
		D5W	24 h
	(16 g/240 mL) 133 mg/mL	NS	24 h
		D5W	24 h

* NS: normal saline; ** D5W: 5% dextrose.

**Table 6 antibiotics-14-00466-t006:** List of antibiotics and compounds used for solution preparation.

	Trade Name and Manufacturer	Batch Number
Amoxicillin	Amoxicilline PANPHARMA 1 g (La Selle-en-Luitré, France)	308350
Cefepime	Céfépime PANPHARMA 1 g Céfépime PANPHARMA 2 g	V2–15 V1–11
Cefiderocol	Céfiderocol 1 g—FETCROJA^®^ (Osaka, Japan)	23K02857
Cefotaxime	Céfotaxime VIATRIS 2 g (Canonsburg, PA, USA)	230572
Cefoxitin	Céfoxitine PANPHARMA 1 g Céfoxitine PANPHARMA 2 g	V3–01 V4–04
Ceftazidime	Ceftazidime VIATRIS 1 g Ceftazidime VIATRIS 2 g	230657 230861
Cloxacillin	Cloxacilline PANPHARMA	308573
Oxacillin	Oxacilline 1 g—ISTOPEN^®^ (Garching bei München, Germany)	2322901
Piperacillin	Pipéracilline PANPHARMA 4 g	308105
Pyridine	Merck (Darmstadt, Germany)	K54632028

**Table 7 antibiotics-14-00466-t007:** Preparation of test solutions.

Antibiotic	Amount/Volume	Solvent
	Concentration	
Amoxicillin	(3 g/240 mL) 12.5 mg/mL	NS *
	(6 g/240 mL) 25 mg/mL	NS
	(12 g/240 mL) 50 mg/mL	NS
Cefepime	(3 g/240 mL) 12.5 mg/mL	NS
		D5W **
Cefiderocol	(3 g/240 mL) 12.5 mg/mL	NS
		D5W
	(6 g/240 mL) 25 mg/mL	NS
		D5W
Cefotaxime	(3 g/240 mL) 12.5 mg/mL	NS
		D5W
Cefoxitin	(3 g/240 mL) 12.5 mg/mL	NS
		D5W
Ceftazidime	(3 g/240 mL) 12.5 mg/mL	NS
		D5W
	(6 g/240 mL) 25 mg/mL	NS
		D5W
Cloxacillin	(6 g/240 mL) 25 mg/mL	NS
		D5W
	(12 g/240 mL) 50 mg/mL	NS
		D5W
Oxacillin	(6 g/240 mL) 25 mg/mL	NS
		D5W
	(12 g/240 mL) 50 mg/mL	NS
		D5W
Piperacillin	(12 g/240 mL) 50 mg/mL	NS
		D5W
	(16 g/240 mL) 133 mg/mL	NS
		D5W

* NS: normal saline; ** D5W: 5% dextrose.

## Data Availability

The original contributions presented in this study are included in the article/Appendix A. Further inquiries can be directed to the corresponding author(s).

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
