# Peer review of "Stability of Nine Time-Dependent Antibiotics for Outpatient Parenteral Antimicrobial Therapy (OPAT) Use"

_antibiotics, 2025, doi:10.3390/antibiotics14050466_

Round 1
Reviewer 1 Report
Comments and Suggestions for Authors
This study evaluates the stability of nine time-dependent antibiotics in elastomeric diffusers at 32°C, providing crucial data for optimizing outpatient parenteral antimicrobial therapy (OPAT). The manuscript is well-structured, methodologically sound, and addresses a clinically relevant gap in the literature. The findings have significant implications for OPAT protocols, particularly in reducing nursing visits and improving patient compliance. However, minor revisions are recommended to enhance clarity, statistical rigor, and clinical applicability.
Minor Concerns
- The manuscript mentions ANOVA and Cochran tests but does not provide detailed statistical outputs (e.g., p-values) for key comparisons. Please include these to strengthen validity.
- Clarify whether interday/intraday precision was assessed for all antibiotics or only for oxacillin/cloxacillin.
- While pyridine levels are reported for ceftazidime, a brief discussion of safe thresholds (e.g., Jones et al.’s 100 mg/day limit) would help contextualize the clinical risk.
- Acknowledge potential real-world variability in device temperature (e.g., patient mobility, ambient conditions) and suggest mitigations (e.g., isothermal bags).
- Explicitly state that results are specific to silicone elastomers and may not apply to polyisoprene devices.
- Note that stability in syringes (used in ICU) was not evaluated, which could be a focus for a follow-up study.
- Please check for typos throughout the text, particularly regarding the phrase "PK/PD data has confirmed."
- Ensure all abbreviations (e.g., NS, D5W) are defined at their first use in the text.
Please review the text for any typos.
Author Response
Comment 1 : The manuscript mentions ANOVA and Cochran tests but does not provide detailed statistical outputs (e.g., p-values) for key comparisons. Please include these to strengthen validity.
Response 1 : We have added in Table 1, the p-values : p<0.05.
Comment 2 : Clarify whether interday/intraday precision was assessed for all antibiotics or only for oxacillin/cloxacillin.
Response 2 : The inter- and intraday precision had been validated in our first published work by Loeuille et al. But given the time lag between that work published in 2022 and our new stability data which is the subject of this manuscript, we have revalidated the inter- and intraday precision for all the molecules evaluated. The results are presented in Table 1 of the manuscript.
Comment 3 : While pyridine levels are reported for ceftazidime, a brief discussion of safe thresholds (e.g., Jones et al.’s 100 mg/day limit) would help contextualize the clinical risk.
Response 3 : In the discussion section, the risks in the event of pyridine overdose are indicated with three references: « Pyridine is considered to be a central nervous system depressant in humans and can cause liver, kidney and digestive disorders. »
A sentence has been added : « Given the doses of pyridine measured in our stability study, the risk seems us minimal. »
[15] French National Institute of Research and Safety for the Prevention of Occupational Accidents and Diseases. 2011. INRS toxicological card of pyridine (FT 85). INRS, Paris, France.
[16] International Agency for Research on Cancer (IARC). 2005. Pyridine. IARC, Lyon, France. [Internet]. [cited mai 2021]. Available at http://www.inchem.org/documents/iarc/vol77/77 -16.html.
We have modified this reference by a more recent article :
[16] IARC Monographs on the Evaluation of Carcinogenic Risks to Humans. Some chemicals that cause tumours of the urinary tract in rodents. Pyridine. IARC, Lyon, France, 2019, Volume 119, pp. 173-203.
[17] Jones TE, Selby PR, Mellor CS, et al. Ceftazidime stability and pyridine toxicity during continuous i.v. infusion. Am J Health-Syst Pharm. 2019 ;76(4) :200 5.
Comment 4 : Acknowledge potential real-world variability in device temperature (e.g., patient mobility, ambient conditions) and suggest mitigations (e.g., isothermal bags).
Response 4 : We have added a section to the discussion – Temperature section :
« Inside the elastomeric device, the temperature of the solution can vary depending on how the device is placed: protected or not from light, under the patient's clothes or not. The use of an isothermal bag would reduce this temperature variation, but after research with our suppliers of elastomeric dispensers, this type of bag did not exist. »
Comment 5 : Explicitly state that results are specific to silicone elastomers and may not apply to polyisoprene devices.
Response 5 : In the discussion section, a sentence is already presented : « We chose silicone diffusers due to their availability in our hospital. However, we do not recommend extrapolating data from silicone diffusers to apply to polyisoprene diffusers. »
We have added this sentence to be more clear : « The results obtained in this work are only available for the preparation of silicone elastomer dispensers and may not apply to polyisoprene devices. »
Comment 6 : Note that stability in syringes (used in ICU) was not evaluated, which could be a focus for a follow-up study.
Response 6 : We have added this sentence at the beginning of the discussion : « For intensive care units (ICU), stability studies in syringes are necessary and are presented in the work by Loeuille et al. [5]. »
Comment 7 : Please check for typos throughout the text, particularly regarding the phrase "PK/PD data has confirmed."
Response 7 : PK/PD was defined in the line 4 of INTRODUCTION.
Comment 8 : Ensure all abbreviations (e.g., NS, D5W) are defined at their first use in the text.
Response 8 : Done.
Reviewer 2 Report
Comments and Suggestions for Authors
The article provides a thorough evaluation of the chemical stability of antibiotics in silicone elastomeric devices for OPAT (Outpatient Parenteral Antimicrobial Therapy) use, using high-performance liquid chromatography (HPLC). The analysis of data such as linearity, precision, and stability is presented well, but there are several areas where additional information or clarification would strengthen the work. Below are my comments:
- The use of normal saline (NS) and 5% dextrose (D5W) as solvents is mentioned frequently, but no control comparison is provided for the solvents themselves. It would be useful to include control experiments where only the solvents (without antibiotics) are tested to ensure that the changes in pH or visual evaluations are due to the antibiotics and not the solvents.
- The linearity results are presented with very high R² values (close to 1), which suggest excellent calibration, but the precision values (less than 2.5% for most antibiotics) may need further explanation. It is crucial to discuss the significance of such low variance values in the context of long-term storage and clinical variability.
- In Table 2, concentrations at some time points are missing (e.g., For amoxicillin, T8 h and T24 h for the 50 mg/mL NS condition). This inconsistency should be addressed and, if necessary, explained in the discussion.
- The presence of pyridine (Table 3) is an interesting result. However, the article doesn’t explain why pyridine is present and how it may influence the stability of ceftazidime. There is also no mention of any potential risk or toxicity related to pyridine accumulation, which would be crucial for clinical application.
- While stability is assessed well, the potential toxicity of breakdown products or reactions with the elastomeric devices is not addressed. The authors should discuss whether degradation products may pose a risk to patients. This would be particularly relevant for antibiotics administered over extended periods, as in OPAT.
- The changes in pH for certain antibiotics (e.g., oxacillin, cefoxitin) over time should be further explained in the discussion. While some pH variations are within an acceptable range, others exceed one unit, which could potentially affect the stability and efficacy of the drugs. How does the pH change correlate with the observed antibiotic degradation? The authors should discuss the potential impact of pH variation on drug stability and therapeutic efficacy.
- The prestudy (lines 263-272) is well described, but the selection criteria for which antibiotics were tested in elastomeric devices versus glass vials could be explained more clearly. Why were cefiderocol, cloxacillin, and oxacillin excluded initially?
- The validation process follows ICH guidelines, but details on LOD (Limit of Detection), LOQ (Limit of Quantification), and accuracy are missing. These are essential aspects of method validation and should be included.
Some Minor issues:
- The description for Table 2 incorrectly includes content related to Table 3, likely due to a formatting or copy-paste error. Please revise the table legends to ensure that each table is described accurately and independently.
- The term "antibiotics stability antibiotics" (line 260) seems like a typographical error.
- Several references have spacing/line breaks/typos (e.g., “Antibioti cs 2025,”, “1052. 347 2.”, “39–354. 355 5.”). Also, Inconsistent punctuation and journal abbreviations (some full, some abbreviated, some mixed).
Overall good.
Author Response
Comment 1 : The use of normal saline (NS) and 5% dextrose (D5W) as solvents is mentioned frequently, but no control comparison is provided for the solvents themselves. It would be useful to include control experiments where only the solvents (without antibiotics) are tested to ensure that the changes in pH or visual evaluations are due to the antibiotics and not the solvents.
Response 1 : Following this revision of the manuscript, in response to this comment, we studied the pH and physical stability (visual and subvisual examination) of 3 diffusers of 0.9% NaCl and 3 diffusers of G5% stored at 32°C for 24 hours. Visual examination was compliant (no change in colour, precipitate or gas formation) for 24 hours, subvisual examination was compliant with European Pharmacopoeia monograph 2.9.19. The change in pH was also less than 0.18 units for 24 hours.
A sentence has been added in the methods of the main document :
« Control experiments
To check that the pH and physical changes are not related to the solvent but to the antibiotic itself, three ED of 0.9% NaCl and three ED of D5W have been prepared and stored at 32°C during 24 hours. pH assessment, visual examination and subvisceral examination were performed for 24 hours. »
And in results :
« Control experiments
About the ED of 0.9% NaCl and D5W alone, visual examination was compliant (no change in colour, precipitate or gas formation) for 24 hours, subvisual examination was compliant with European Pharmacopoeia monograph 2.9.19. The change in pH was also less than 0.18 units for 24 hours. »
Comment 2 : The linearity results are presented with very high R² values (close to 1), which suggest excellent calibration, but the precision values (less than 2.5% for most antibiotics) may need further explanation. It is crucial to discuss the significance of such low variance values in the context of long-term storage and clinical variability.
Response 2 : Intra-day and inter-day precision was achieved by preparing three different solutions each day. A small variability in the preparation of the solutions may explain the precision of less than 1.92% for intra-day precision and less than 3.79 for inter-day precision, which seems acceptable to us and usually accepted in stability studies of solutions published in the literature.
Comment 3 : In Table 2, concentrations at some time points are missing (e.g., For amoxicillin, T8 h and T24 h for the 50 mg/mL NS condition). This inconsistency should be addressed and, if necessary, explained in the discussion.
Response 3 : The choice of analysis times was made on the basis of several criteria: (1) in collaboration with an infectiologist, (2) taking into account human constraints (difficulties in carrying out analyses at T12h), (3) time constraints (duration of sample analysis by HPLC sometimes does not allow analysis to be carried out at T6h and T8h), (4) in relation to results obtained at lower times (e.g. instability of amoxicillin at 50 mg/mL at T8h, so an additional analysis was limited to T24h): instability). Analyses at T0, T8h, T12h and T24h were preferred in order to carry out a maximum of 1 to 3 administrations per day and therefore a maximum of 1 to 3 nursing visits.
Comment 4 : The presence of pyridine (Table 3) is an interesting result. However, the article doesn’t explain why pyridine is present and how it may influence the stability of ceftazidime. There is also no mention of any potential risk or toxicity related to pyridine accumulation, which would be crucial for clinical application.
Response 4 : In the method section, we have added a sentence to explain why pyridine is present : « The ceftazidime molecule has a pyridine ring which is released during hydrolysis. It is one of the major degradation products of ceftazidime but is considered toxic. »
In the discussion section, there is a specific section for pyridine. The risks in the event of pyridine overdose are indicated with three references: « Pyridine is considered to be a central nervous system depressant in humans and can cause liver, kidney and digestive disorders. »
A sentence has been added : « Given the doses of pyridine measured in our stability study, the risk seems us minimal. »
[15] French National Institute of Research and Safety for the Prevention of Occupational Accidents and Diseases. 2011. INRS toxicological card of pyridine (FT 85). INRS, Paris, France.
[16] International Agency for Research on Cancer (IARC). 2005. Pyridine. IARC, Lyon, France. [Internet]. [cited mai 2021]. Available at http://www.inchem.org/documents/iarc/vol77/77 -16.html.
We have modified this reference by a more recent article :
[16] IARC Monographs on the Evaluation of Carcinogenic Risks to Humans. Some chemicals that cause tumours of the urinary tract in rodents. Pyridine. IARC, Lyon, France, 2019, Volume 119, pp. 173-203.
[17] Jones TE, Selby PR, Mellor CS, et al. Ceftazidime stability and pyridine toxicity during continuous i.v. infusion. Am J Health-Syst Pharm. 2019 ;76(4) :200 5.
Comment 5 : While stability is assessed well, the potential toxicity of breakdown products or reactions with the elastomeric devices is not addressed. The authors should discuss whether degradation products may pose a risk to patients. This would be particularly relevant for antibiotics administered over extended periods, as in OPAT.
Response 5 : To the best of our knowledge, no publication has reported a container/content interaction between the antibiotics studied and the elastomeric devices. We have not found any publications concerning the release of plasticisers in elastomer devices. So, we have not considered this aspect in this publication.
We have added a sentence in the publication : « For each stability study, we have searched the additional peaks present on the chromatograms at each analysis time. Their evolution during the analyses was determined as well as their presence in the analysis of a quality sample. The percentage of AUC of additional peak/AUC of antibiotic of interest was measured for each antibiotic and was less than 2% for each antibiotic. Additional peaks that are present after preparation (T0) and evolve constantly over time, or additional peaks that are also present on the chromatogram of the CRS analysis, are considered impurities. A literature search for potential degradation products (DP) for each antibiotic was also carried out. To our knowledge, with the exception of ceftazidime with pyridine, none study demonstrated the toxicity of a PD for an antibiotic studied in this study. »
Comment 6 : The changes in pH for certain antibiotics (e.g., oxacillin, cefoxitin) over time should be further explained in the discussion. While some pH variations are within an acceptable range, others exceed one unit, which could potentially affect the stability and efficacy of the drugs. How does the pH change correlate with the observed antibiotic degradation? The authors should discuss the potential impact of pH variation on drug stability and therapeutic efficacy.
Response 6 : There is not necessarily a correlation between variations in pH and variations in the concentration of a molecule of interest measured by HPLC. The pH is measured on a solution containing: an active ingredient, excipients, solvent and a container (permeability).
Solutions with a pH variation of more than 1 unit were a limiting factor for stability and therefore not recommended for administration to the patient.
Comment 7 : The prestudy (lines 263-272) is well described, but the selection criteria for which antibiotics were tested in elastomeric devices versus glass vials could be explained more clearly. Why were cefiderocol, cloxacillin, and oxacillin excluded initially?
Response 7 : For cefiderocol, because of the cost of the vial, we did not want to carry out a pre-study. For cloxacillin and oxacillin, there are human and material limitations (difficulty in accessing these two products during other pre-studies), so we decided that for these two molecules we would start the stability study straight away.
As recommanded by guidelines of SFPO/ESOP (Bardin, C, Astier A, Vulto A, Sewell G, Vigneron J, Trittler R, Daouphars M, Paul M, Trojniak M, Pinguet F, et al. Guidelines for the Practical Stability Studies of Anticancer Drugs: A European Consensus Conference. Ann. Pharm. Fr. 2011, 69, 221–231.), pre-studies in small glass vials need small volume of products and are less expensive.
Comment 8 : The validation process follows ICH guidelines, but details on LOD (Limit of Detection), LOQ (Limit of Quantification), and accuracy are missing. These are essential aspects of method validation and should be included.
Response 8 :
- LOD/LOQ
Limit of quantification (LOQ) determination is not necessary in the case of a stability study the objective of which is to quantify near 100% of antibiotic concentration.
But in methods : a sentence has been added : « Accuracy has been validated for each method and values of limit of detection (LOD) an limit of quantification (LOQ) are presented in Table 1. »
And Results of LOD/LOQ have added in Table 1.
- Accuracy : Methods : Analytical method validation
A sentence has been added : « For accuracy, three different solutions of three concentrations were prepared three times a day for 3 days for each antibiotics. Accuracy was determined as the difference between the mean measured value and the accepted true value. For each antibiotic, recovery at three concentration levels must be between 98 and 102% to validate accuracy.»
And Results of accuracy has added in Table 1.
Some Minor issues:
Comment 9 : The description for Table 2 incorrectly includes content related to Table 3, likely due to a formatting or copy-paste error. Please revise the table legends to ensure that each table is described accurately and independently.
Response 9 : We have modified the sentence.
Comment 10 : The term "antibiotics stability antibiotics" (line 260) seems like a typographical error.
Response 10 : We have modified the sentence
Comment 11 : Several references have spacing/line breaks/typos (e.g., “Antibioti cs 2025,”, “1052. 347 2.”, “39–354. 355 5.”). Also, Inconsistent punctuation and journal abbreviations (some full, some abbreviated, some mixed).
Response 11 : Done.
Round 2
Reviewer 2 Report
Comments and Suggestions for Authors
The revised manuscript addresses the majority of reviewer’s comments thoroughly and constructively. Overall, the manuscript has been significantly improved and is suitable for publication.